# Antibiotic discovery with artificial intelligence for the treatment of *Acinetobacter baumannii* infections

Yassir Boulaamane,[1] Irene Molina Panadero,[2] Abdelkrim Hmadcha,[3,4] Celia Atalaya Rey,[2] Soukayna Baammi,[5] Achraf El Allali,[5] Amal Maurady,[1,6] Younes Smani[2,3]

**ABSTRACT** Global challenges presented by multidrug-resistant *Acinetobacter baumannii* infections have stimulated the development of new treatment strategies. We reported that outer membrane protein W (OmpW) is a potential therapeutic target in *A. baumannii*. Here, a library of 11,648 natural compounds was subjected to a primary screening using quantitative structure-activity relationship (QSAR) models generated from a ChEMBL data set with >7,000 compounds with their reported minimal inhibitory concentration (MIC) values against *A. baumannii* followed by a structure-based virtual screening against OmpW. *In silico* pharmacokinetic evaluation was conducted to assess the drug-likeness of these compounds. The ten highest-ranking compounds were found to bind with an energy score ranging from −7.8 to −7.0 kcal/mol where most of them belonged to curcuminoids. To validate these findings, one lead compound exhibiting promising binding stability as well as favorable pharmacokinetics properties, namely demethoxycurcumin, was tested against a panel of *A. baumannii* strains to determine its antibacterial activity using microdilution and time-kill curve assays. To validate whether the compound binds to the selected target, an OmpW-deficient mutant was studied and compared with the wild type. Our results demonstrate that demethoxycurcumin in monotherapy and in combination with colistin is active against all *A. baumannii* strains. Finally, the compound was found to significantly reduce the *A. baumannii* interaction with host cells, suggesting its anti-virulence properties. Collectively, this study demonstrates machine learning as a promising strategy for the discovery of curcuminoids as antimicrobial agents for combating *A. baumannii* infections.

**IMPORTANCE** *Acinetobacter baumannii* presents a severe global health threat, with alarming levels of antimicrobial resistance rates resulting in significant morbidity and mortality in the USA, ranging from 26% to 68%, as reported by the Centers for Disease Control and Prevention (CDC). To address this threat, novel strategies beyond traditional antibiotics are imperative. Computational approaches, such as QSAR models leverage molecular structures to predict biological effects, expediting drug discovery. We identified OmpW as a potential therapeutic target in *A. baumannii* and screened 11,648 natural compounds. We employed QSAR models from a ChEMBL bioactivity data set and conducted structure-based virtual screening against OmpW. Demethoxycurcumin, a lead compound, exhibited promising antibacterial activity against *A. baumannii*, including multidrug-resistant strains. Additionally, demethoxycurcumin demonstrated anti-virulence properties by reducing *A. baumannii* interaction with host cells. The findings highlight the potential of artificial intelligence in discovering curcuminoids as effective antimicrobial agents against *A. baumannii* infections, offering a promising strategy to address antibiotic resistance.

**KEYWORDS** *Acinetobacter baumannii*, antimicrobial resistance, QSAR modeling, molecular modeling, antibacterial assays

Address correspondence to Younes Smani, ysma@upo.es.

The authors declare no conflict of interest.

See the funding table on p. 14.

10.1128/msystems.00325-24 **1**

Antimicrobial resistance (AMR) in Gram-negative bacteria (GNB) has become a serious problem in recent years, with potentially devastating impacts on the economy and human life (1). The need for more effective and safer antimicrobial compounds has become increasingly urgent in the post-antibiotic era (1). *Acinetobacter baumannii*, one of the six superbug *Enterococcus faecium, Staphylococcus aureus, Klebsiella pneumoniae, Acinetobacter baumannii, Pseudomonas aeruginosa,* and *Enterobacter spp.* (ESKAPE) pathogens, is a global priority pathogen for the development of effective antimicrobial therapies, due to rapid changes in the genetic constitution of *A. baumannii* and the plasticity to acquire different resistance mechanisms (2–4). The scarce development of efficient antibiotics against this microorganism has sparked renewed scientific interest in finding effective antimicrobial agents capable of killing, inhibiting growth, or inhibiting the activity of essential virulence factors of *A. baumannii* (5).

The extensive functions of outer membrane proteins (OMPs) in GNB have led to their identification as potential drug targets (6). Among the OMPs, outer membrane protein W (OmpW) is a porin playing a pivotal role in the uptake of nutritional substances such as iron (7). Several studies have highlighted the relevance of OmpW as a potential drug target in GNB. For instance, researchers investigated how *A. baumannii* adapts to low oxygen conditions during infection. They found that OmpW was downregulated in hypoxic conditions. To understand its role as a virulence factor, they studied the effects of OmpW loss in *A. baumannii*. They discovered that the absence of OmpW reduced *in vitro* the bacterium's ability to adhere to and invade host cells, to cause cell death, and to form biofilm without affecting its growth and *in vivo* the pathogenicity of *A. baumannii* (8). Similarly, *Vibrio cholerae* mutant strains lacking OmpW showed reduced colonization in the mouse intestine compared with strains expressing OmpW (9). The collective evidence from these studies strongly suggests that OmpW plays a crucial role in bacterial pathogenesis and could be a promising target for the development of drugs aimed at combating GNB infections.

Natural products have long been a subject of great interest in the development of novel antimicrobial drugs (10). These products, derived from plants, animals, and microorganisms, have been used for centuries by various traditional medicine systems to treat infections (11).

Chemical libraries enable comprehensive virtual drug screening by offering a diverse range of compounds. Large databases enhance the integration of advanced methods like machine learning and artificial intelligence for accurate prediction of drug properties. For example, Massachussetts Institute of Technology (MIT) researchers used artificial intelligence to identify a potent new antibiotic known as halicin. This compound demonstrates efficacy against a wide range of bacteria, including some that exhibit resistance to all known antibiotics. Furthermore, halicin displayed no significant side effects in mice, prompting researchers to plan further development and clinical trials (12). Recently discovered by researchers at the University of Toronto in 2021, abaucin exhibits promising efficacy against the lethal superbug *A. baumannii*. Although still in early development, it holds significant potential in the treatment of drug-resistant infections (13).

Thus, the objective of the present study was to screen a large library of natural products with potential activity against *A. baumannii* using "*in silico*" and "*in vitro*" assays. The screening focused on compounds targeting the function of OmpW. A library of 11,648 natural compounds was retrieved from Ambinter chemical library, and an *in silico* approach combining data-driven and molecular modeling methods for drug discovery was employed. Artificial-based quantitative-structure activity relationship (QSAR) models were developed to predict the bioactivity of the natural products against *A. baumannii*. The retained compounds were subsequently subjected to molecular docking screens and absorption, distribution, metabolism, and excretion (ADME) evaluation to assess their pharmacological and pharmacokinetic profiles. The best compounds, which exhibited a strong affinity for OmpW along with favorable pharmacokinetic properties, were further evaluated through molecular dynamics simulations. Finally, a

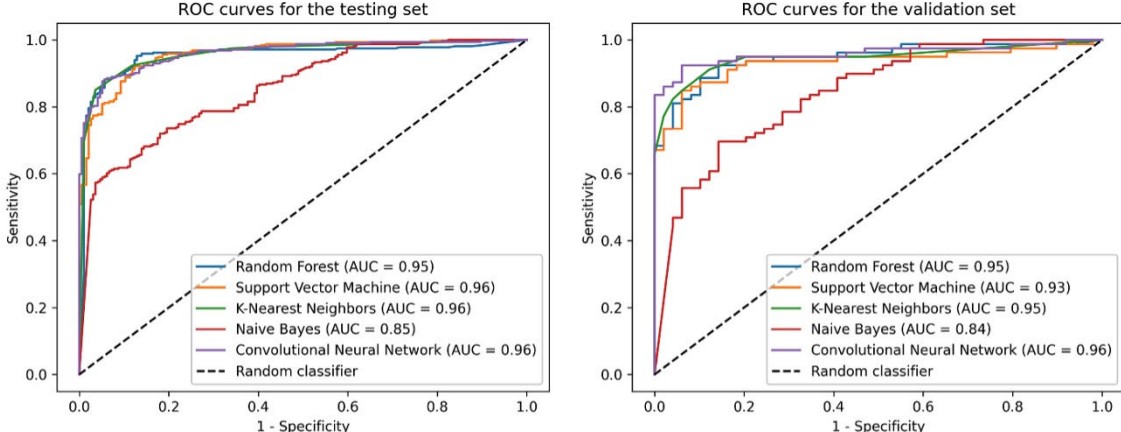

**FIG 1**  Performance of QSAR classification models on test and validation sets. The ROC curve and AUC values illustrate model performance. The CNN model resulted as the best classifier in the test and validation sets. QSAR: quantitative structure-activity relationship; ROC: receiver-operating characteristic; AUC: area under the curve; CNN: convolutional neural network.

lead candidate was subjected to *in vitro* testing to assess its potential for inhibiting *A. baumannii* growth.

## RESULTS

### QSAR screening

In this study, four machine learning algorithms known for their efficacy in QSAR modeling were chosen: random forest, support vector machine, k-nearest neighbors, and Gaussian naïve Bayes, based on previous reports of their performance (14). Furthermore, we have developed a convolutional neural network (CNN) using sequential architecture consisting of embedding, convolutional, pooling, flattened, and dense layers. Beginning with an embedding layer mapping input data to dense vectors, the model subsequently utilizes two convolutional layers with rectified linear unit (ReLU) activation for feature extraction, followed by max-pooling layers for dimensionality reduction. The flattened output is fed into dense layers, facilitating non-linear transformations and classification. With a final sigmoid activation layer for binary classification, the model is trained using binary cross-entropy loss and Adam optimizer, aiming to discern compound properties efficiently for screening active compounds against *A. baumannii*. The hyperparameters of the four machine learning classifiers underwent optimization using the GridSearchCV module within Scikit-Learn (v1.2.2) (15).

The performance of the QSAR classification models was evaluated using area under the curve (AUC) scores, with all models demonstrating excellent AUC values, as depicted

**TABLE 1**  Performance metrics of the generated classification models on the testing and validation sets[a]

| Data set | Model | SE | SP | Q+ | Q− | ACC | F1 score | MCC |
|---|---|---|---|---|---|---|---|---|
| Testing set | Random forest | 0.89 | 0.92 | 0.88 | 0.94 | 0.91 | 0.89 | 0.82 |
| | Support vector machine | 0.88 | 0.91 | 0.85 | 0.93 | 0.90 | 0.91 | 0.78 |
| | k-nearest neighbors | 0.88 | 0.93 | 0.88 | 0.92 | 0.91 | 0.92 | 0.80 |
| | Naïve Bayes | 0.58 | 0.96 | 0.96 | 0.56 | 0.72 | 0.72 | 0.53 |
| | Convolutional neural network | 0.83 | 0.96 | 0.95 | 0.88 | 0.90 | 0.92 | 0.81 |
| Validation set | Random forest | 0.86 | 0.91 | 0.86 | 0.91 | 0.89 | 0.91 | 0.77 |
| | Support vector machine | 0.81 | 0.91 | 0.86 | 0.87 | 0.87 | 0.89 | 0.72 |
| | k-nearest neighbors | 0.79 | 0.96 | 0.94 | 0.85 | 0.88 | 0.90 | 0.77 |
| | Naïve Bayes | 0.53 | 0.93 | 0.94 | 0.49 | 0.66 | 0.64 | 0.45 |
| | Convolutional neural network | 0.81 | 0.97 | 0.96 | 0.86 | 0.90 | 0.91 | 0.80 |

[a]SE: sensitivity (true positive rate); SP: specificity (false positive rate); Q+: positive predictive value (precision); Q−: negative predictive value; ACC: accuracy; MCC: Matthews' correlation coefficient.

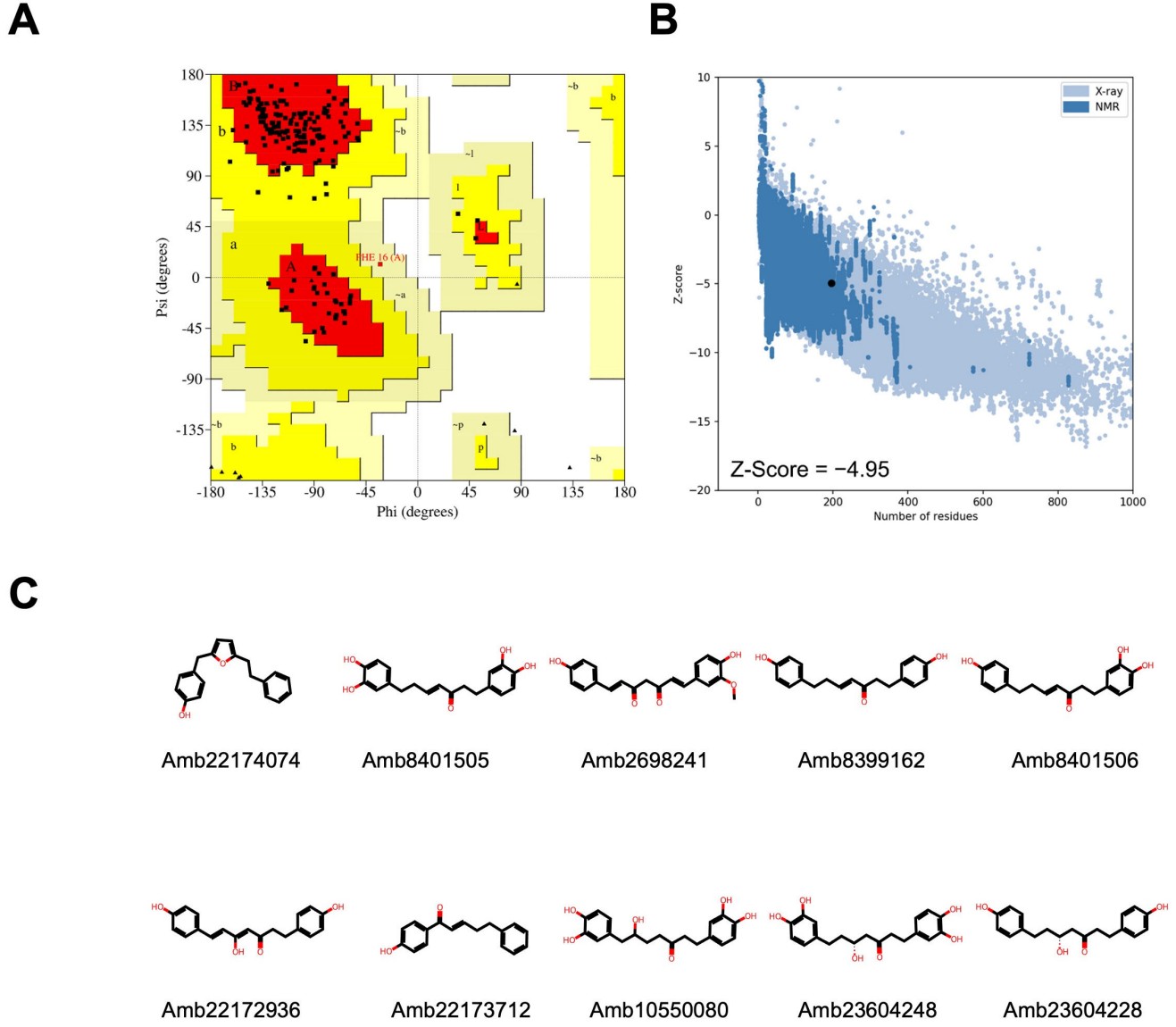

**FIG 2** Ramachandran plot for OmpW displaying the distribution of each amino acid within the favored, allowed, and disallowed regions (A). Scatter plot and Z-score revealing the overall model quality of OmpW (B). Chemical structures of the top ten highest-scoring compounds against OmpW. OmpW: outer membrane protein W (C).

in Fig. 1. Furthermore, the performance of QSAR models was assessed using various metrics such as precision, F1 score, accuracy, and Matthews correlation coefficient (MCC), as outlined in Table 1. Notably, the CNN model exhibited excellent performance on both the test and validation sets, achieving an AUC value of 0.96. Consequently, it was chosen to predict the activity of Ambinter drug-like natural compounds by comparing their molecular descriptors with those in the training data set and leveraging the learned relationships. At this step, 1,193 compounds were identified as active against *A. baumannii* and subsequently selected for the structure-based virtual screening study.

## Docking screens of natural products

The quality assessment of the AlphaFold model of OmpW (Uniprot ID: A0A335FU53) (16, 17), according to Ramachandran plot, shows that 92.2% of residues are in the most favorable regions, 7.2% in allowed regions, 0.6% in generously disallowed regions, and 0.0% in disallowed regions. Validation of the OmpW structure using Protein Structure

Analysis-web (ProSA-web) shows a Z-score value of −4.95, which is within the range of scores typically found for native proteins of similar size (Fig. 2A and B). The predicted active compounds were subjected to molecular docking screens, and their binding affinities were ranked accordingly. Specifically, we observed that the highest-ranking compounds exhibit binding scores ranging from −7.0 to −7.8 kcal/mol and belong to curcuminoids as shown in Fig. 3. The amino acids involved in the ligand binding are presented in Table 2.

Docking poses of the highest-ranking compounds are displayed in Fig. 3. In brief, the structural analysis of the docked compounds reveals consistent hydrogen bond formation between the hydroxyl (-OH) group of the phenyl ring in curcuminoids and the amino acid residue GLN-23. Furthermore, we detected additional hydrogen bond interactions implicating key residues, namely ASN-104, THR-109, and LYS-195, situated within the periplasmic site of OmpW. Additionally, our analysis reveals multiple instances of hydrophobic interactions, with notable involvement of amino acid residues PHE-59, HIS-101, ASN-144, and GLN-146.

## ADME evaluation

A significant proportion, approximately 40%, of drug candidates fail during clinical trials primarily due to inadequate ADME properties (18). *In silico* ADME prediction offers a rapid method to assess the drug-likeness of a compound by calculating its physicochemical properties. This approach substantially reduces the time and resources required during the overall drug development process. In this study, SwissADME

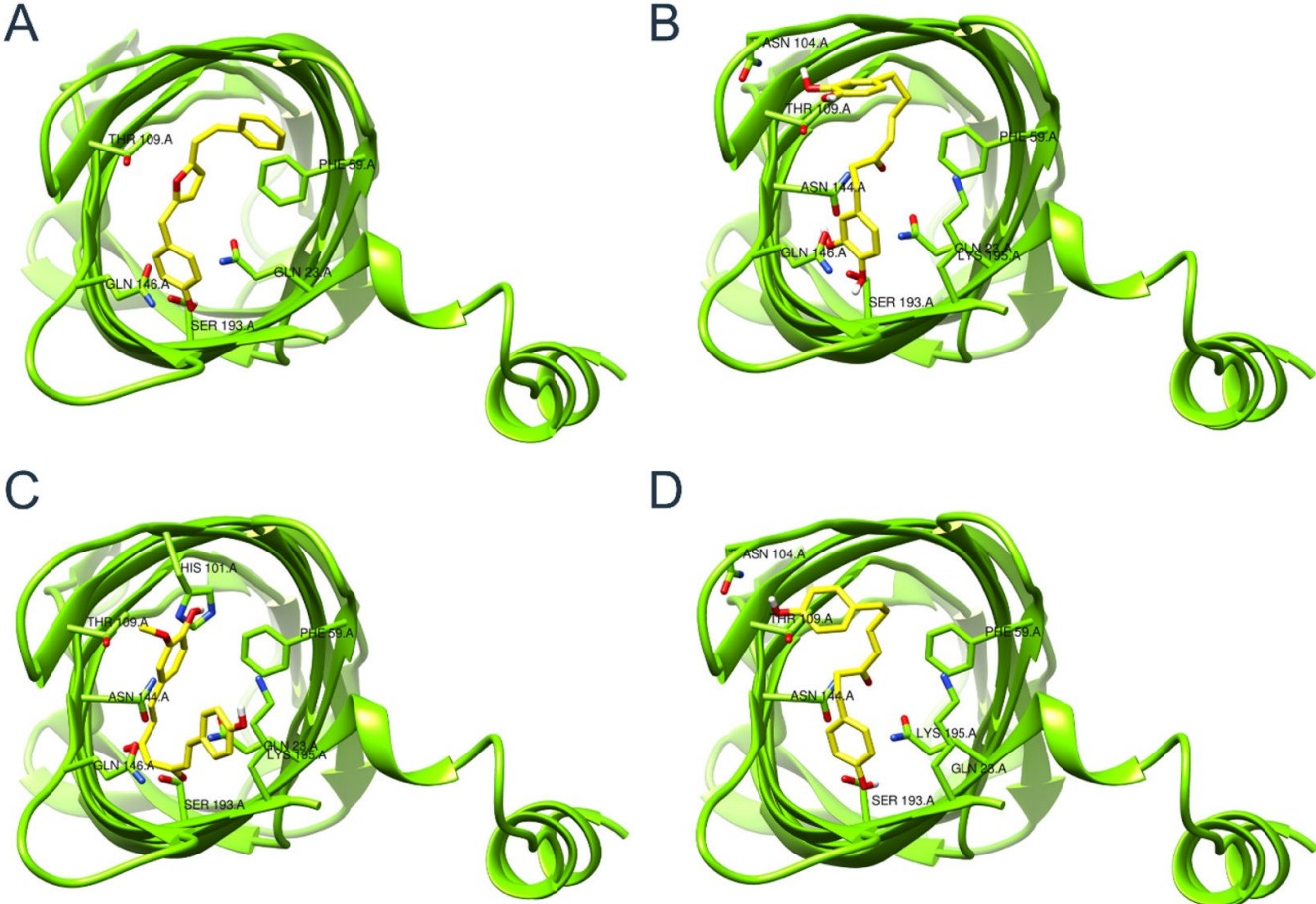

**FIG 3** Binding conformations of the top four highest-ranking natural products: Amb22174074 (A), Amb8401505 (B), Amb2698241 (C), and Amb8399162 (D) in complex with OmpW's periplasmic region. OmpW: outer membrane protein W.

**TABLE 2** Structure-based virtual screening results of the selected natural compounds against OmpW of *A. baumannii*

| Compound | Binding score (kcal/mol) | Hydrogen bonds | Hydrophobic interactions |
|---|---|---|---|
| Amb22174074 | −7.8 | GLN-23, SER-193, LYS-195 | PHE-59, HIS-101, ASN-144, GLN-146, LYS-195 |
| Amb8401505 | −7.7 | GLN-23, PHE-102, ASN-104, ASN-144, TRP-153, SER-193 | PHE-59, HIS-101, LYS-103, ASN-144, LYS-195 |
| Amb2698241 | −7.5 | GLN-23, HIS-101, SER-193, LYS-195 | PHE-59, THR-109, ASN-144, GLN-146, LYS-195 |
| Amb8399162 | −7.4 | GLN-23, ASN-104, SER-193, LYS-195 | PHE-59, LYS-103, THR-109, ASN-144, LYS-195 |
| Amb8401506 | −7.4 | PHE-102, ASN-104, GLN-146, LYS-195 | PHE-59, HIS-101, LYS-103, LYS-195 |
| Amb22172936 | −7.4 | GLN-23, ARG-107, THR-109, TRP-153, LYS-195 | HIS-101, LYS-103, THR-109, ASN-144, GLN-146, LYS-195 |
| Amb22173712 | −7.4 | GLN-23, THR-109, SER-193, LYS-195 | PHE-59, HIS-101, LYS-103, THR-109, ASN-144, GLN-146, LYS-195 |
| Amb10550080 | −7.3 | GLN-23, ASN-104, THR-109, ASN-152, SER-193 | PHE-59, HIS-101, LYS-195 |
| Amb23604248 | −7.2 | ASN-104, THR-109, ASN-144, GLN-146, SER-193 | PHE-59, LYS-103, THR-109 |
| Amb23604228 | −7.0 | GLN-23, ASN-104, ASN-144, SER-193, LYS-195 | GLN-23, ASN-104, ASN-144, SER-193, LYS-195 |

(http://www.swissadme.ch/) was employed to compute various pharmacokinetic properties of the highest-scoring compounds to evaluate their drug-likeness and suitability for further experimental studies (19). ADME properties for the selected compounds are shown in Table 3. The results reveal that all the compounds possess good lipophilicity in accordance with Lipinski's rule of five; moreover, water solubility values were found to be in the recommended range for most drugs. Intestinal absorption was found to be high in all the compounds. Of the top 10 compounds tested for blood-brain barrier (BBB) permeability, only five were found to be unable to penetrate the BBB. This is a crucial finding, as antibacterial compounds should not exert their effects on the central nervous system (CNS). None of the compounds were found to act as a P-glycoprotein substrate; thus, their bioavailability is not impacted by this protein. Finally, the pan-assay interference compounds (PAINS) test has revealed four compounds presenting one alert in their structure due to the presence of the catechol group, which can result in non-specific binding with various target proteins.

## Molecular dynamics simulations and binding free energy

In the molecular docking study, the protein structure was treated as rigid. To gain deeper insights into the protein-ligand interactions, molecular dynamics simulations were performed on the docked complexes in a water environment for 100 ns. The root-mean square deviation (RMSD) was measured relative to the OmpW structure bound to the selected candidates. Fig. 4A illustrates the protein RMSD values for the top four complexes, showing a consistently stable RMSD of 0.3 nm during most of the simulation, except for Amb22174074, which displayed higher fluctuations exceeding 0.3 nm in the last 20 ns. The analysis of the ligand RMSD showed values between 0.1 and 0.25 nm for most ligands, suggesting minor conformational changes during the

**TABLE 3** ADME properties' prediction results for the selected compounds

| Compound | LogS | GI absorption | BBB | P-gp substrate | Bioavailability score | PAINS |
|---|---|---|---|---|---|---|
| Amb22174074 | −4.88 | High | Yes | No | 0.55 | 0 alert |
| Amb8401505 | −3.73 | High | No | No | 0.55 | Catechol_A |
| Amb2698241 | −3.92 | High | No | No | 0.56 | 0 alert |
| Amb8399162 | −4.01 | High | Yes | No | 0.55 | 0 alert |
| Amb8401506 | −3.87 | High | Yes | No | 0.55 | Catechol_A |
| Amb22172936 | −4.17 | High | Yes | No | 0.55 | 0 alert |
| Amb22173712 | −4.02 | High | Yes | No | 0.55 | 0 alert |
| Amb10550080 | −3.11 | High | No | No | 0.55 | Catechol_A |
| Amb23604248 | −3.11 | High | No | No | 0.55 | Catechol_A |
| Amb23604228 | −3.39 | High | Yes | No | 0.55 | 0 alert |

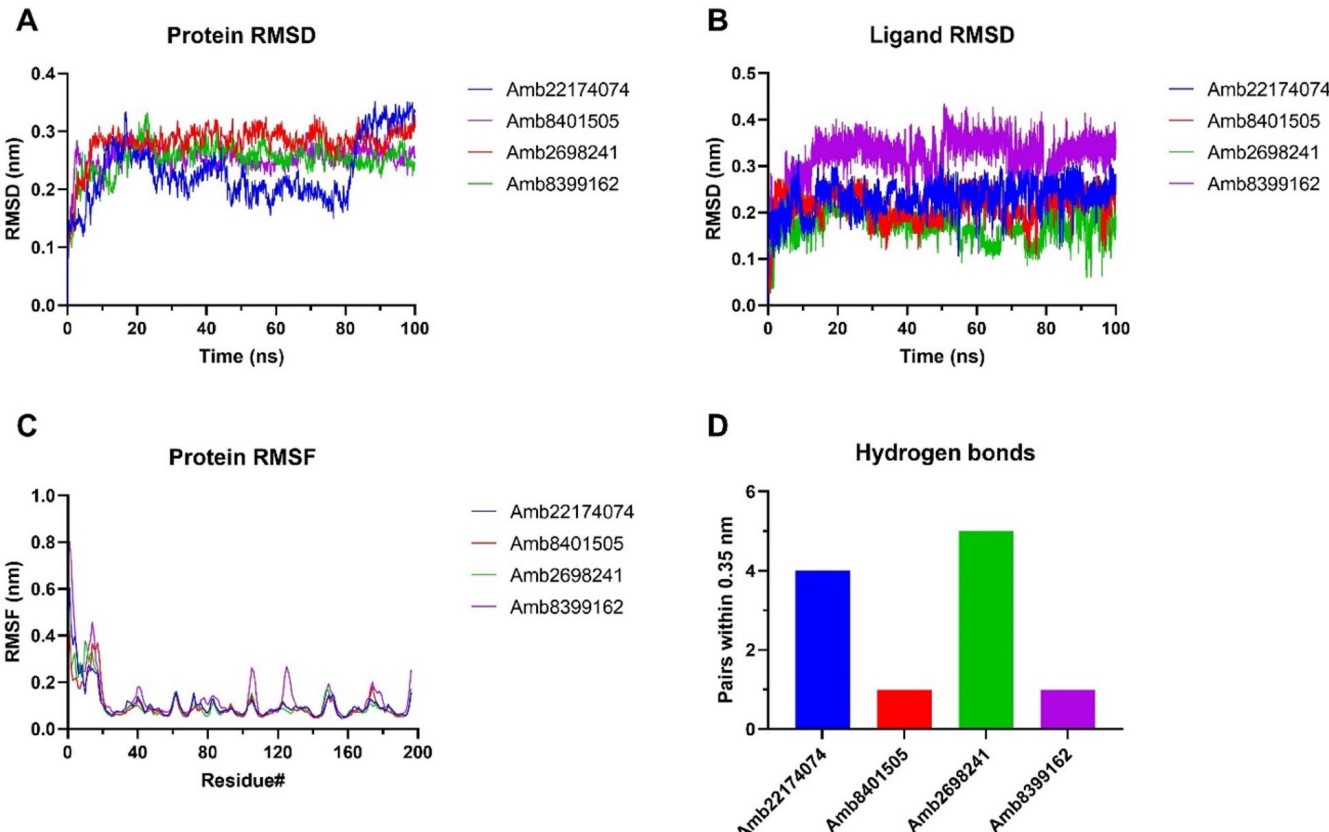

**FIG 4** Molecular dynamics simulations analysis through protein RMSD (A), ligand RMSD (B), RMSF (C), and hydrogen bonds at 100 ns (D).

simulation. However, the ligand Amb8399162 deviated from this trend, with an RMSD of 0.35 nm, suggesting a more significant conformational change (Fig. 4B). In Fig. 5C, the graph illustrates the variations observed in each amino acid. Notably, the N-terminal region exhibited the highest fluctuations, which is a common characteristic. For all other residues, minor fluctuations of approximately 0.1 nm were observed, except for Amb8399162, which displayed fluctuations higher than 0.2 nm in certain regions of the periplasm. Finally, hydrogen bonds within a proximity of 0.35 nm were documented. Fig. 4D depicts the hydrogen bonds observed at 100 ns, with Amb2698241 forming four hydrogen bonds, highlighting its stable and consistent binding to the protein. The average free binding energy of the selected complexes was determined using the g_mmpbsa package (v1.6) (20, 21).

The binding energy was computed by combining the scores of Van der Waals energy, electrostatic energy, polar solvation, and SASA energy as presented in Table 4. The highest binding energy was observed in Amb2698241 (−45.23 kJ/mol), suggesting a strong binding to the target protein.

## Antibacterial activity

The best compound exhibiting the lowest docking score as well as favorable ADME properties was demethoxycurcumin (Amb2698241). The MIC was then assessed using microdilution assays against different reference *A. baumannii* ATCC 17978 strains, its isogenic mutant deficient in OmpW, and colistin-resistant *A. baumannii* clinical isolates. Demethoxycurcumin inhibited bacterial growth at a concentration of 64 mg/L for all the studied strains (Table 5).

Colistin potentiation is critical for safeguarding this last resort antibiotic as it is often our only treatment option against highly resistant Gram-negative pathogens.

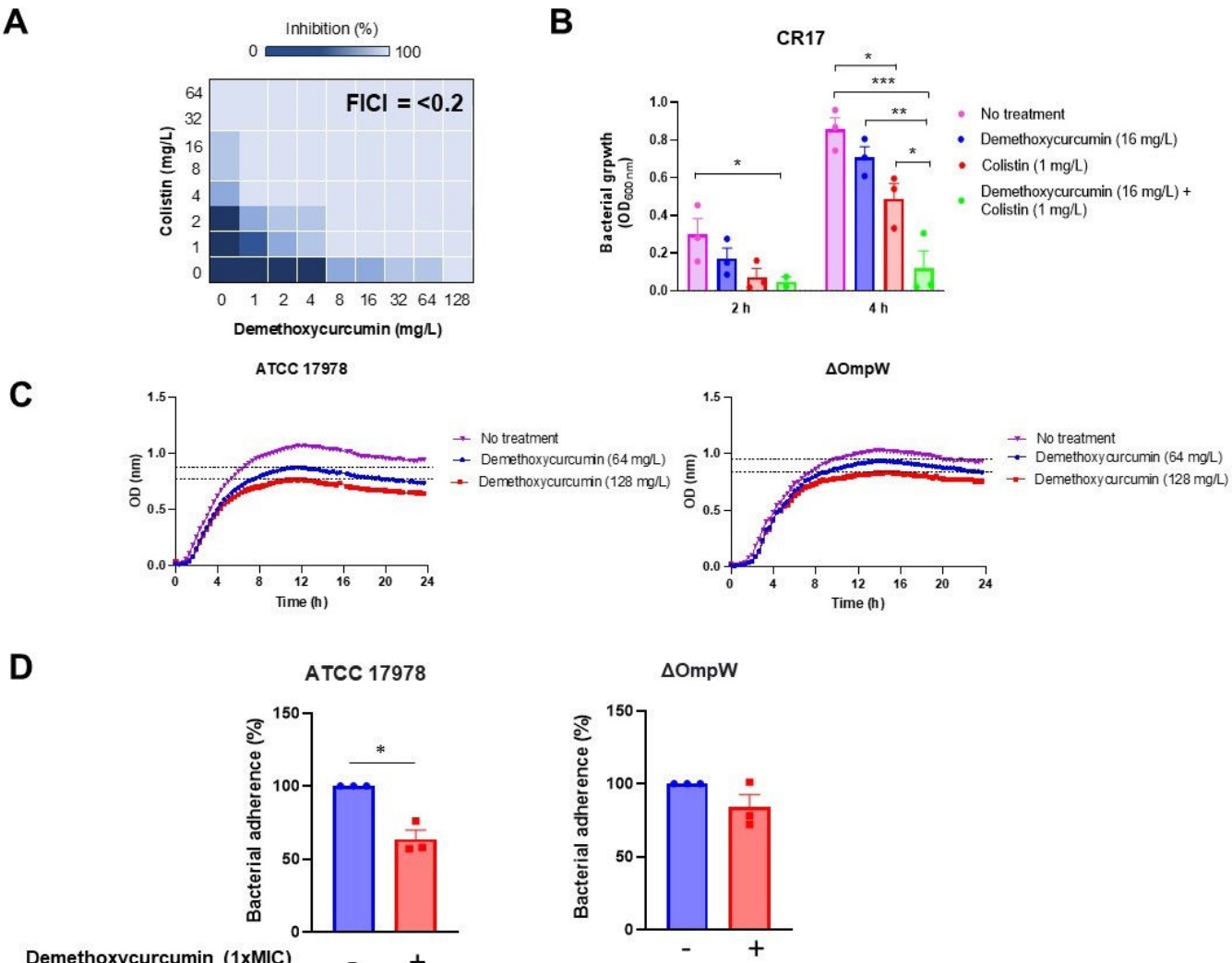

FIG 5 *In vitro* antibacterial activity of demethoxycurcumin. Representative heat plots of microdilution checkerboard assay for the combination of colistin and demethoxycurcumin against colistin-resistant *A. baumannii* CR17 strain (A). Bacterial growth for colistin and demethoxycurcumin monotherapy and combination therapy against colistin-resistant *A. baumannii* CR17 strain during 24 h incubation. The concentrations of colistin and demethoxycurcumin are 1 and 16 mg/L, respectively. The data are presented as means ± standard errors of the means (SEM), and analysis of variance (ANOVA) test followed by the post hoc Tukey test was used for statistical analysis. *$P < 0.05$: colistin vs no treatment, demethoxycurcumin plus colistin vs no treatment, and demethoxycurcumin plus colistin vs colistin, **$P < 0.01$: demethoxycurcumin plus colistin vs demethoxycurcumin, ***$P < 0.001$: demethoxycurcumin plus colistin vs no treatment (B). Bacterial growth curve plots of *A. baumannii* ATCC 17978 and *A. baumannii* ΔOmpW in the absence and presence of demethoxycurcumin treatment at different concentrations (C). Analysis of *A. baumannii* ATCC 17978 and ΔOmpW adhesion to HeLa host cells with and without demethoxycurcumin treatment. The data are presented as means ± SEM, and student *t*-test was used for statistical analysis. *$P < 0.05$: treatment vs no treatment (D).

We examined whether demethoxycurcumin can sensitize colistin-resistant clinical strain CR17. Checkerboard assay showed that demethoxycurcumin at ≥1 mg/L demonstrated synergy with colistin against CR17 strain. Demethoxycurcumin at ≥8 mg/L in

TABLE 4 List of average and standard deviations of all energetic components including the binding energy taken from MM-PBSA analysis

| Complex | MMPBSA (kJ/mol) | | | | |
|---|---|---|---|---|---|
| | $\Delta G_{bind}$ | $\Delta G_{vdW}$ | $\Delta G_{elec}$ | $\Delta G_{solv}$ | $\Delta G_{sasa}$ |
| Amb22174074 | −35.03 ± 20.08 | −118.74 ± 16.41 | −45.52 ± 24.70 | 144.30 ± 33.25 | −15.10 ± 1.63 |
| Amb8401505 | −41.92 ± 17.02 | −122.42 ± 15.16 | −42.96 ± 15.32 | 139.20 ± 22.25 | −15.74 ± 1.58 |
| Amb2698241 | −45.23 ± 17.96 | −115.48 ± 18.37 | −37.25 ± 11.57 | 122.31 ± 23.21 | −14.81 ± 1.54 |
| Amb8399162 | −39.11 ± 16.56 | −143.59 ± 18.84 | −35.11 ± 16.60 | 156.61 ± 30.64 | −17.01 ± 1.94 |

**TABLE 5**  MIC results for the studied compounds against different wild-type, colistin-resistant, and OmpW-deficient *A. baumannii*

| *A. baumannii* strain | MIC (mg/L) | |
|---|---|---|
| | Colistin | Demethoxycurcumin |
| ATCC 17978 | 0.25 | 64 |
| ATCC17978 ΔOmpW | 0.25 | 64 |
| Ab11 | 256 | 64 |
| Ab20 | 64 | 64 |
| Ab21 | 128 | 64 |
| Ab22 | 128 | 64 |
| Ab99 | 64 | 64 |
| Ab113 | 256 | 64 |
| CR17 | 32 | 64 |

combination with colistin increased the activity of colistin against CR17 strain, with a fractional inhibitory concentration index (FICI) of <0.2 (Fig. 5A). In addition, the combination between 16 mg/L demethoxycurcumin and 1 mg/L colistin exhibited a synergistic effect during 2 and 4 h, reducing the bacterial growth compared with colistin demethoxycurcumin alone (Fig. 5B).

Using bacterial growth assays, we examined the antibacterial activity of demethoxy-curcumin against ATCC 17978 and ΔOmpW strains. Figure 5C reveals that *A. baumannii* ATCC 17978 exhibits rapid growth, reaching 0.5 OD within the first 4 h. However, a noticeable disparity in growth is observed between the control sample and the samples treated with demethoxycurcumin, particularly at higher compound concentrations (2× MIC and 4× MIC). A similar trend of growth inhibition is observed in the ΔOmpW strain, although it demonstrates a higher OD value compared with *A. baumannii* ATCC 17978 in the presence of demethoxycurcumin treatment. This disparity in growth can be attributed to the resistance of the mutant strain to the compound, as the absence of OmpW may hinder the compound's ability to exert its effect, as indicated by the findings of the molecular docking study.

In addition, and to evaluate the effect of demethoxycurcumin on *A. baumannii* interaction with host cells, we studied the adherence of ATCC 17978 and ΔOmpW strains to HeLa cells for 2 h in the presence of demethoxycurcumin. Treatment with demethoxy-curcumin at 1× MIC reduced the adherence of ATCC 17978 and ΔOmpW strains to HeLa cells by 36% and 16%, respectively (Fig. 5D).

## DISCUSSION

In this study, we present a multi-stage approach for screening bioactive compounds from extensive databases. This approach combines data-driven QSAR models and structure-based virtual screening methods for drug discovery. Our classification models demonstrated strong performance in distinguishing between active and inactive compounds, achieving AUC values ranging from 0.85 to 0.96 for the testing set and 0.84 to 0.96 for the validation set. The results of molecular docking indicated binding affinities spanning from −5.4 to −7.8 kcal/mol. Notably, the top-scoring compounds belong to the curcuminoid chemical class, recognized for their antibacterial activities (22, 23).

Analysis of molecular interactions revealed a consistent hydrogen bond formation with GLN-23 in most of the compounds under study. Additional hydrophobic interactions involved the following amino acids: PHE-59, HIS-101, ASN-144, and GLN-146. Molecular dynamics analysis of the first four complexes displayed remarkable stability throughout the simulation, except for the tricyclic compound Amb22174074, which exhibited some deviations, leading to an RMSD of 0.3 nm. This observation could be attributed to the inherent limited flexibility of this compound, prompting conformational changes in the protein.

Furthermore, our investigation identified van der Waals energy as the primary contributor to the stability of the complexes, as determined by the MMPBSA method. To

validate our *in silico* results, we assessed a lead candidate, demethoxycurcumin, for its *in vitro* activity in monotherapy and in combination with colistin against an extensive range of *A. baumannii* strains, including colistin-resistant strains. This lead candidate presents an antibacterial activity as shown by microdilution and time-kill curve assays. Notably, a reduction in compound activity against OmpW-deficient mutant has been observed in the time-kill curve assay. Li et al. showed that demethoxucurcumin present antibacterial activity in monotherapy and in combination with gentamicin against another pathogen, methicillin-resistant *Staphylococcus aureus* (24)

Our findings suggest the crucial role of the OmpW in facilitating the compound's activity. Previous studies reported the binding of colistin and tamoxifen metabolites to OmpW (25, 26).

Bacterial adhesion to and invasion into host cells are important steps in causing *A. baumannii* infection (27). It is well-known that OmpW plays a key role in host-pathogen interactions. Deletion of OmpW reduced *A. baumannii*'s adherence and invasion into host cells, as well as its cytotoxicity (8). Similarly, in the absence of OprG, which is homologous to OmpW in *P. aeruginosa*, this pathogen was significantly less cytotoxic against human bronchial epithelial cells (28). OmpW is essential for *A. baumannii* to disseminate between organs and to cause the death of mice, as observed for other pathogens such as *V. cholerae* (9). Motley et al. reported an increase in OmpW expression during *E. coli* infection in a murine granulomatous pouch model (29), and OmpW has been shown to protect *E. coli* against host responses, conferring resistance to complement-mediated killing and phagocytosis (30, 31). All these previous studies indicated that OmpW could be a potential drug target in GNB to develop new treatments. However, no data have been reported on the effect of natural products on host-*A. baumannii* interactions. To our knowledge, this study provides the first evidence for the effect of demethoxycurcumin in reducing *A. baumannii*'s adherence to host cells. Moreover, this effect is consistent with time-kill curve data. Further studies are needed, such as animal infection models, to validate the potential use of demethoxycurcumin as monotherapy and in combination with antibiotics used in clinical settings.

In summary, this study demonstrated a multi-step computational and experimental approach to identify natural products as potential therapeutics targeting the OmpW protein of *A. baumannii*. Demethoxycurcumin was validated as an active lead compound both *in vitro* and in reducing bacterial interaction with host cells. Further investigations are necessary, such as testing in animal models of infection, to validate the therapeutic potential of targeting OmpW by demethoxycurcumin and related natural products.

## MATERIALS AND METHODS

### QSAR modeling

A bioactivity data set from the ChEMBL database, which comprised the chemical structures of 11,014 compounds along with their reported MIC values against *A. baumannii*, was acquired (32). To ensure the reliability of the data, the data set by only keeping those with MIC values of the same unit (mg/L) was carefully curated. For duplicate compounds with multiple reported activities, a mean value was calculated, and only one entry was kept in the study using the Pandas (v2.2.0) library in Python (33). The processed data set consisted of 3,196 compounds. To classify the compounds, molecules with reported MIC values < 32 were labeled as active, whereas molecules with MIC >64 were labeled as inactive. This resulted in 1,310 active compounds and 816 inactive compounds. For further analysis, the RDKit cheminformatics suite (v2023.09.4) was used to generate 2,048 bits of molecular descriptors using Morgan fingerprints (34, 35). These descriptors were derived from the compounds' Simplified Molecular-Input Line-Entry System (SMILES) representation and were based on the widely used extended-connectivity fingerprints (ECFP4) (36).

To support the training and assessment of our QSAR models, we partitioned the data sets into a unified train/test set (80%) and a distinct validation set (20%). Within the

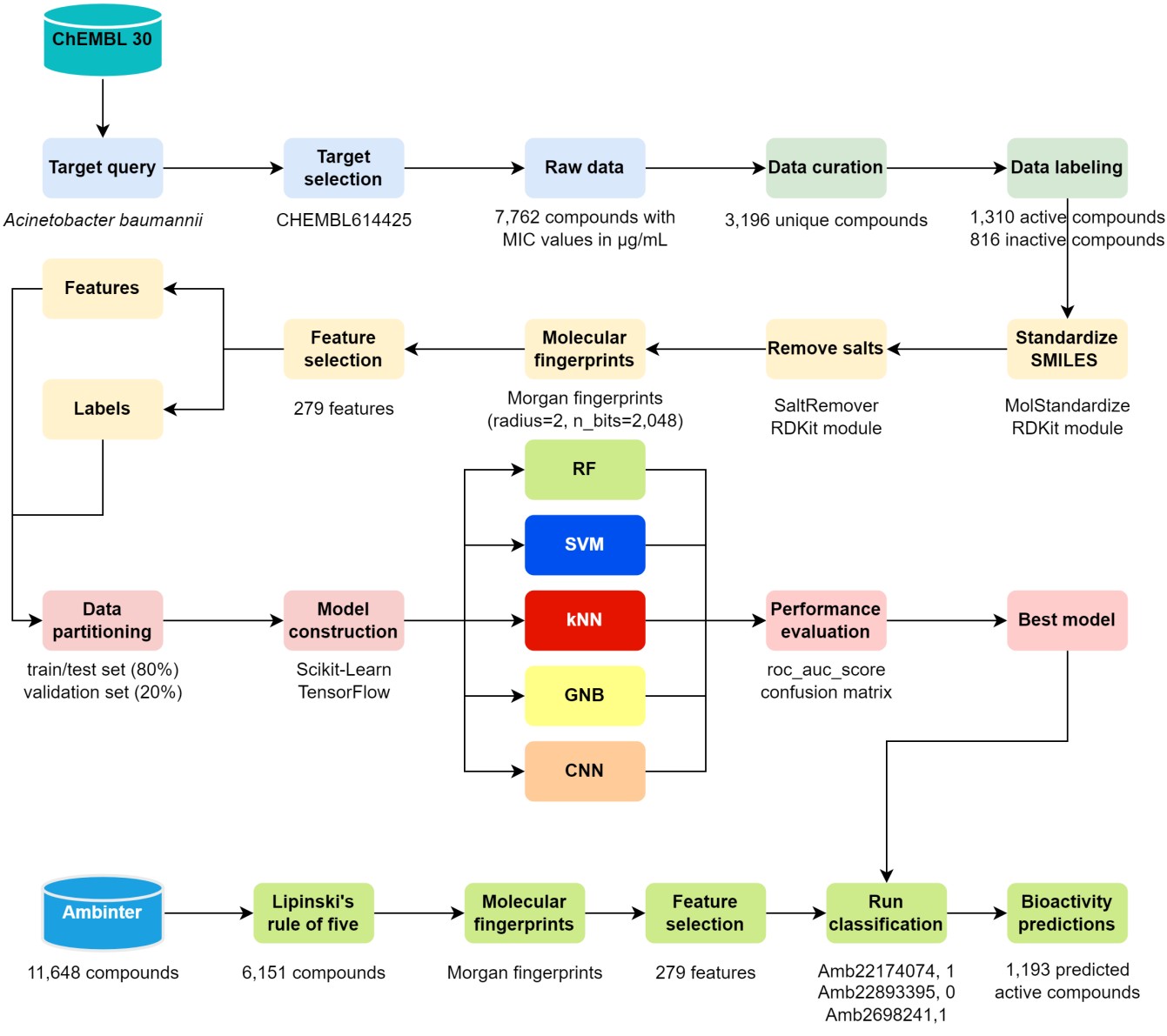

**FIG 6** The QSARBioPred workflow outlines the processes involved in constructing QSAR models aimed at predicting the likelihood of a compound being active against *A. baumannii*. This involves generating molecular fingerprints of the compounds and employing machine learning techniques to discern patterns correlated with activity. Subsequently, the model enables the screening of novel compounds for potential activity against *A. baumannii*.

train/test set, an 80/20 split further divided the data into train and test subsets for model training and evaluation, respectively. The validation set was exclusively allocated for the final evaluation of the selected model's performance on unseen data, as depicted in Fig. 6. A standard workflow for our proposed QSAR approach, along with the source code, can be found in the GitHub repository (https://github.com/yboulaamane/QSARBioPred/).

## Protein structure preparation

To refine and enhance the quality of the 3D protein model, the online server GalaxyRefine (https://galaxy.seoklab.org/cgi-bin/submit.cgi?type=REFINE) was used (37). The platform employs a multi-step approach that involves side-chain rebuilding, side-chain repacking, and molecular dynamics simulation to achieve overall structure relaxation. Subsequently, the PROCHECK algorithm was employed through the SAVES webserver

([https://saves.mbi.ucla.edu/](https://saves.mbi.ucla.edu/)) (38) to generate Ramachandran plots, whereas ProSA-web was used to assess model accuracy and statistical significance using a knowledge-based potential (39).

## Binding site detection

The plausible binding pockets for the selected OmpW protein structure were predicted using PrankWeb ligand binding site prediction webserver ([https://prankweb.cz/](https://prankweb.cz/)) (40). Fig. S1 depicts the 3D structure of OmpW with their predicted binding pockets shown as residues with different colors. The predicted binding pockets scores, grid coordinates, and residue IDs are shown in Table S1.

## Structure-based virtual screening

The natural compounds were retrieved from Ambinter natural compounds library ([https://www.ambinter.com/](https://www.ambinter.com/)). Eleven thousand six hundred forty-eight compounds were evaluated for their drug-likeness by computing their physicochemical properties such as molecular weight, LogP, number of hydrogen bond donors/acceptors, and the number of rotatable bonds DataWarrior (v06.01.00) (41). According to Lipinski's rule of five, only 6,151 compounds were retained for further analysis (42). Structure-based virtual screening was performed using AutoDock Vina (v1.1.2) with a Perl script to automate the molecular docking process as published in our previous study (43, 44). The 3D structure of OmpW was optimized using AutoDockTools (v1.5.6) by adding polar hydrogens and computing Kollman charges (45). The grid box was centered around the coordinates provided by PrankWeb for the best-scoring pockets. The pocket (2) located near the periplasmic of the β-barrel structure was selected for molecular docking as mentioned in the literature (46).

Docking snapshots were generated using UCSF Chimera 1.17.3 (47). Molecular interactions were visualized using Protein-Ligand Interaction Profiler ([https://plip-tool.biotec.tu-dresden.de/plip-web/plip/index](https://plip-tool.biotec.tu-dresden.de/plip-web/plip/index)) (48).

## Molecular dynamics simulations and binding free energy calculation

Molecular dynamics simulations were performed using GROMACS (v2019.3) (49, 50) to evaluate the stability of selected candidates in complexes with OmpW. The CHARMM36 force field generated the protein topology file, whereas the CGENFF server assigned parameters to ligands (51). TIP3P water model solvated the protein-ligand systems in a cubic box, with Na+ and Cl− ions added for charge neutrality. To optimize the energy, the steepest descent technique was employed, setting Fmax not to exceed 1,000 kJ/mol/nm. Subsequently, two consecutive 1 ns simulations using canonical constant number of molecules, volume and temperature (T) (NVT) and isobaric constant number of molecules, pressure and temperature (NPT) ensembles were performed to equilibrate the systems at 300 Kelvin and 1 bar pressure. All simulations were conducted under periodic boundary conditions (PBC), and long-range electrostatic interactions were handled using the particle mesh Ewald method (52). For data collection, 100 ns molecular dynamics simulations were conducted. To analyze the dynamic behavior of the selected complexes, various geometric properties such as root-mean-square deviation (RMSD), root-mean-square fluctuation (RMSF), and hydrogen bonds were calculated using GROMACS (v2019.3) (53).

The binding free energies of the screened complexes were calculated using the molecular mechanics Poisson–Boltzmann surface area (MM-PBSA) method (54). The binding free energy ($\Delta E_{\text{binding}}$) is determined using the following equations:

$$\Delta E_{\text{binding}} = E_{\text{complex}} - \left( E_{\text{inhibitor}} + E_{\text{OmpW}} \right) \tag{1}$$

Equation 1 is the total MMPBSA energy of the protein-ligand complex, where $E_{OmpW}$ and $E_{inhibitor}$ are the isolated proteins and ligands' total free energies in solution, respectively.

$$\Delta G_{binding} = \Delta G_{vdW} + \Delta G_{elec} + \Delta G_{solv} + \Delta G_{sasa} \qquad (2)$$

Equation 2 defines the generalized MMPBSA as the sum of four energies: electrostatic ($\Delta G_{elec}$), van der Waals ($\Delta G_{vdw}$), polar ($\Delta G_{solv}$), and SASA ($\Delta G_{sasa}$).

## Antibacterial activity assays

### Microdilution assay

The MIC of demethoxycurcumin was determined against ATCC 17978 strain, an isogenic mutant deficient in OmpW, and seven colistin-resistant *A. baumannii* clinical strains, along with 24 clinical strains, in two independent experiments using the broth microdilution method, in accordance with the standard guidelines of the European Committee on Antimicrobial Susceptibility Testing (EUCAST) (55). A $5 \times 10^5$ CFU/mL inoculum of each strain was cultured in Luria Bertani (LB) and added to U bottom microtiter plates (Deltlab, Spain) containing demethoxycurcumin. The plates were incubated for 18 h at 37°C.

### Bacterial growth curve assay

To determine the antibacterial activity, bacterial growth curves of the ATCC 17978 strain and its isogenic deficient in OmpW (ΔOmpW) and CR17 strain were performed in triplicate in 96-well plate (Deltlb, Spain). An initial inoculum of $5 \times 10^5$ CFU/mL was prepared in LB in the presence of 1× MIC, 2× MIC, and 4× MIC of demethoxycurcumin. A drug-free broth was evaluated in parallel as a control. Plates were incubated at 37°C with shaking, and bacterial growth was monitored for 24 h using a microtiter plate reader (Tecan Spark, Austria).

### Checkerboard assay

The assay was performed on a 96-well plate in duplicate as previously described (56). Colistin was 2-fold serially diluted along the *x* axis, whereas demethoxycurcumin was 2-fold serially diluted along the *y* axis to create a matrix, where each well consists of a combination of both agents at different concentrations. Bacterial cultures grown overnight were then diluted in saline to 0.5 McFarland turbidity, followed by 1:50 further dilution LB and inoculation on each well to achieve a final concentration of approximately $5.5 \times 10^5$ CFU/mL. The 96-well plates were then incubated at 37°C for 18 h and examined for visible turbidity. The fractional inhibitory concentration (FIC) of the colistin was calculated by dividing the MIC of colistin in the presence of demethoxycurcumin by the MIC of colistin alone. Similarly, the FIC of demethoxycurcumin was calculated by dividing the MIC of demethoxycurcumin in the presence of colistin by the MIC of rafoxanide alone. The FIC index was the summation of both FIC values. FIC index values of ≤0.5 were interpreted as synergistic.

## Human cell culture

HeLa cells were grown in 24-well plates in Dulbecco's Modified Eagle medium (DMEM) supplemented with 10% heat-inactivated fetal bovine serum (FBS), vancomycin (50 mg/L), gentamicin (20 mg/L), amphotericin B (0.25 mg/L) (Invitrogen, Spain), and 1% 4-(2-hydroxyethyl)-1-piperazineethanesulfonic acid (HEPES) in a humidified incubator with 5% $CO_2$ at 37°C. HeLa cells were routinely passaged every 3 or 4 days. Immediately before infection, HeLa cells were washed three times with prewarmed phosphate buffered saline (PBS) and further incubated in DMEM without FBS and antibiotics (57).

## Adhesion assay

HeLa cells were infected with $1 \times 10^8$ CFU/mL of *A. baumannii* ATCC 17978 and ΔOmpW strains in the absence and presence of 1× MIC of demethoxycurcumin at a multiplicity of infection (MOI) of 100 for 2 h with 5% $CO_2$ at 37°C. Subsequently, infected HeLa cells were washed five times with prewarmed PBS and lysed with 0.5% Triton X-100. Diluted lysates were plated onto LB agar (Merck, Spain) and incubated at 37°C for 24 h for enumeration of developed colonies and then the determination of the number of bacteria that attached to HeLa cells (8). All experiments were performed in triplicate.

## Statistical analysis

The GraphPad Prism 9 (version 9.3.1; GraphPad Software, LLC.) statistical package was used. Group data are presented as bar plots and means ± standard errors of the means (SEM). To determine differences between means, an ANOVA test followed by the *post hoc* Tukey test and Student *t*-test was used for the bacterial growth assay and the adherence/invasion assay, respectively. *P* value of $<0.05$ was considered significant.

## ACKNOWLEDGMENTS

This work was co-funded by the Consejería de Universidad and Investigación e Innovación de la Junta de Andalucía (grant ProyExcel_00116) and funded by the Instituto de Salud Carlos III, Subdirección General de Redes y Centros de Investigación Cooperativa, Ministerio de Economía, and Industria y Competitividad (grant PI19/01009), cofinanced by the European Development Regional Fund (A way to achieve Europe, Operative Program Intelligent Growth 2014 to 2020). Y.B. is supported by ERASMUS+ Student Mobility for Traineeships.

Conceptualization: Y.B., A.M., and Y.S.; Methodology: Y.B., I.M.P., A.H., C.A.R., S.B., and A.E.A.; Investigation: Y.B. and Y.S.; Visualization: Y.B. and I.M.P.; Supervision: Y.S. and A.M.; Writing-original draft: Y.B. and Y.S.; Writing-review & editing: Y.B. and Y.S.

## AUTHOR AFFILIATIONS

[1]Laboratory of Innovative Technologies, National School of Applied Sciences of Tangier, Abdelmalek Essaadi University, Tetouan, Morocco

[2]Centro Andaluz de Biología del Desarrollo, Universidad Pablo de Olavide/CSIC/Junta de Andalucía, Seville, Spain

[3]Departamento de Biología Molecular e Ingeniería Bioquímica, Universidad Pablo de Olavide, Seville, Spain

[4]Biosanitary Research Institute (IIB-VIU), Valencian International University (VIU), Valencia, Spain

[5]Bioinformatics Laboratory, College of Computing, Mohammed VI Polytechnic University, Benguerir, Morocco

[6]Faculty of Sciences and Techniques of Tangier, Abdelmalek Essaadi University, Tetouan, Morocco

## AUTHOR ORCIDs

Abdelkrim Hmadcha  http://orcid.org/0000-0002-4105-3702
Younes Smani  http://orcid.org/0000-0001-9302-8384

## FUNDING

| Funder | Grant(s) | Author(s) |
| --- | --- | --- |
| Consejería de Universidad, Investigación e Innovación de la Junta de Andalucía | ProyExcel_00116 | Younes Smani |
| Instituto de Salud Carlos III | PI19/01009 | Younes Smani |

## DATA AVAILABILITY

The data that support the findings of this study are available from the corresponding author upon reasonable request.

## ADDITIONAL FILES

The following material is available online.

### Supplemental Material

**Figure S1 (mSystems00325-24-S0001.pdf).** 3D structure and protein sequence of OmpW.
**Table S1 (mSystems00325-24-S0002.pdf).** Predicted binding pockets for OmpW.

### Open Peer Review

**PEER REVIEW HISTORY (review-history.pdf).** An accounting of the reviewer comments and feedback.

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
