## [Reviewer comments · mSystems]

Antibiotic discovery with artificial intelligence for the treatment of *Acinetobacter baumannii* infections

Yassir Boulaamane, Irene Molina Panadero, Abdelkrim Hmadcha, celia Atalaya Rey, Soukayna Baammi, Achraf El Allali, Amal Maurady, and Younes Smani

Corresponding Author(s): Younes Smani, Universidad Pablo de Olavide

Review Timeline:

Submission Date:	March 6, 2024
Editorial Decision:	March 25, 2024
Revision Received:	March 26, 2024
Accepted:	March 27, 2024

Editor: Neha Garg

Reviewer(s): Disclosure of reviewer identity is with reference to reviewer comments included in decision letter(s). The following individuals involved in review of your submission have agreed to reveal their identity: Simone Zuffa (Reviewer #1)

Transaction Report:

DOI: <https://doi.org/10.1128/msystems.00325-24>

Re: mSystems00325-24 (Antibiotic discovery with artificial intelligence for the treatment of *Acinetobacter baumannii* infections)

Dear Dr. Younes Smani:

Thank you for the privilege of reviewing your work. Your manuscript is almost ready for acceptance pending minor textual changes. Below you will find my comments, instructions from the mSystems editorial office, and the reviewer comments. In addition, there are some inconsistencies throughout the text on words that should be in italics or not. I would also suggest not to use "artificial intelligence" in the title. Maybe change it with machine learning.

Revision Guidelines

Sincerely,
Neha Garg
Editor
mSystems

Reviewer #1 (Comments for the Author):

No further comment.

Please just correct how p values are represented in Figure 5B. For convention: * p < 0.05, ** p < 0.01 and *** p < 0.001 and so

on. Please use the appropriate * to represent p values obtained from Tukey's.

Re: mSystems00325-24R1 (Antibiotic discovery with artificial intelligence for the treatment of *Acinetobacter baumannii* infections)

Dear Dr. Younes Smani:

Your manuscript has been accepted, and I am forwarding it to the ASM production staff for publication. Your paper will first be checked to make sure all elements meet the technical requirements. ASM staff will contact you if anything needs to be revised before copyediting and production can begin. Otherwise, you will be notified when your proofs are ready to be viewed.

Cover Image Submissions: If you would like to submit a potential Cover Image, please email a file and a short legend to msystems@asmusa.org. Please note that we can only consider images that (i) the authors created or own and (ii) have not been previously published. By submitting, you agree that the image can be used under the same terms as the published article. Image File requirements: TIF/EPS, 7.5 inches wide by 8.25 inches tall (at least 2,250 pixels wide by 2,475 pixels tall), minimum 300 dpi resolution (600 dpi preferred), RGB, and no figure elements, e.g., arrows or panel labels. The legend should be a short description of the image, 1-2 sentences recommended.

Sincerely,
Neha Garg